# Non-Neuronal Transmitter Systems in Bacteria, Non-Nervous Eukaryotes, and Invertebrate Embryos

**DOI:** 10.3390/biom12020271

**Published:** 2022-02-08

**Authors:** Yuri B. Shmukler, Denis A. Nikishin

**Affiliations:** Lab of the Problems of Regeneration, N. K. Koltzov Institute of Developmental Biology RAS, Moscow 119334, Russia; d.nikishin@idbras.ru

**Keywords:** receptor, embryo, 5-hydroxytryptamine, dopamine, GABA, SNARE complex, bacteria, protozoa, sea urchin

## Abstract

In 1921, Otto Loewi published his report that ushered in the era of chemical transmission of biological signals. January 2021 marked the 90th anniversary of the birth of Professor Gennady A. Buznikov, who was the first to study the functions of transmitters in embryogenesis. A year earlier it was 60 years since his first publication in this field. These data are a venerable occasion for a review of current knowledge on the mechanisms related to classical transmitters such as 5-hydroxytryptamine, acetylcholine, catecholamines, etc., in animals lacking neural elements and prenervous invertebrate embryos.

## 1. Introduction

The discovery of the chemical transmission of intercellular signals [1] was followed by an enormous amount of research in the field, stimulated both by the fundamental importance of this work and by its obvious relevance for medicine. Further, as it most often happens, the rapidly expanding research eventually began to encompass those areas that, at first glance, had no direct relationship to it. Surprisingly, the first data on the presence of acetylcholine (ACh) outside the nervous system, which appeared in the late 1940s, concerned bacteria [2]. Soon, in the early 1950s, the presence of ACh was discovered also in the germ cells and early embryos of sea urchins [3]. As for the functional significance of this, the author made a curious and unlikely suggestion, especially for ACh, that this easily metabolized substance might be stored for the realization of a future function. However, this work must be acknowledged as the starting point for experimental studies of transmitters in early (prenervous) embryogenesis.

Further steps in this field were the experiments which showed that 5-hydroxytryptamine (serotonin, 5-HT) is able to activate the ciliary motility of prenervous larvae of molluscs [4,5], while transmitter antagonists block cleavage divisions in sea urchins [6]. At the same time, the presence of the 5-HT or 5-HT-like substance was demonstrated in the early embryos of sea urchins and loaches [6,7].

These discoveries were first considered absolute heresy by the physiological community and were vehemently opposed by a number of scientists of the time. However, systematic studies in this field did not begin by chance, but were logically based on the ideas of Academician Koshtoyanz and his associates, according to which nervous transmission originates from some ancestral intracellular function of these substances (see [8]).

Later biochemical, electrophysiological, cytological, radiological and finally molecular biological approaches used in addition to the original embryophysiological methods have produced a large amount of data and concepts in the field. Considering that these chemical substances, commonly referred to as intercellular messengers, are also produced in non-neuronal cells and can function as signaling molecules there, the correct term would be “transmitters” and not “neurotransmitters” [9,10,11]. We will use precisely this term in what follows.

The present work is devoted to the analysis of the data collected so far on non-neuronal transmitter mechanisms in organisms which, in their ontogeny, either have no nervous system at all or are prenervous developmental stages of organisms which later form a nervous system.

## 2. Transmitter Systems in Bacteria

The initial data on the presence of ACh in bacteria (see above) subsequently received confirmations. ACh is synthesized by various bacteria, including bacilli and lactobacilli [12,13]. Horiuchi et al. [14] found ACh in *Escnerichia coli*, *Staphylococcus aureus* and *Bacillus subtilis*. The highest level of ACh was found in *B. subtilis*.

First 5 HT was found in *E. coli* cells from the intestine of ascarids [15]. Originally, the question of the origin of biogenic monoamines, which are identical to eukaryotic transmitters in bacteria, was controversial. It was assumed that in parasitic bacteria they might originate at least in part from the host organism, but prokaryotes could also possibly synthesize biogenic amine transmitters through them. Only further studies of the content of 5-HT in the intestinal tissue of *Ascaris suum* indicate that bacterial 5-HT may contribute [16]. Lactobacteria such as *Lactococcus* also produced 5-HT to some extent [17].

Synthesis of endogenous 5-HT, 5-hydroxyindoleacetic acid, noradrenaline (NA) and dopamine (DA) was detected in *B. subtilis* and *E. coli* K-12 by HPLC [18,19]. Micromolar concentrations of these compounds were detected in *E. coli* cells in the early growth phase; their intracellular content decreases after the transition to the late growth phase. Presumably, enzyme systems corresponding to those in animals are involved in the biosynthesis and degradation of monoamine transmitters in bacterial cells. Micromolar concentrations of DOPA and nanomolar of 5-HT, DA and NA during the late growth phase in the culture fluid of *E. coli* are sufficient for receptor binding [19].

There is a large amount of data on the physiological effects of transmitters in bacteria. The earliest data on this subject date from the early 1990s. The ability of catecholamines to increase the growth of the bacteria *E. coli*, *Yersinia enterocolitica*, and *Pseudomonas aeruginosa* was demonstrated. The growth of *Y. enterocolitica* and *E. coli* could be increased almost 100,000-fold by the addition of NA into the medium in a concentration-dependent manner [20]. DA also increases the number of solitary cells, while other tested amines trigger the formation of cell groups [21]. The effectiveness of these compounds on the growth of Enterobacteriaceae and Pseudomonadaceae families ranges as follows: NA >>> Adr ≥ DA > DOPA [22]. No α- or β-adrenergic receptor agonists or antagonists affected bacterial growth in a dose-dependent manner comparable to that of NA. Also, (-)- NA was more potent than its (+)-enantiomer, as is the case in higher organisms [22]. Both the α-adrenergic receptor antagonist benextramine and the β-antagonist alprenolol have no effect *per se*, nor do they diminish the effect of (-)- NA on *E. coli* growth [22]. Here and below, it should be taken into account that the specificity of the ligands of mammalian transmitter receptors differs significantly from that of non-nervous organisms and prenervous embryos due to very significant differences in the structure of receptors. This is especially true for bacteria, where the nature of transmitter receptors is fundamentally different from that of eukaryotes. In addition, the ligands of transmitter-gated ion channels, in principle, affect the processes of cell division to a much lesser extent than the ligands of metabotropic G-protein coupled receptors.

Later, the effect of serotonergic drugs was discovered. Administration of 5-HT into the medium accelerates the growth of bacterial cultures and induces the aggregation of *E. coli* and *Rhodospirillum rubrum* [23]. In addition, 5-HT and histamine, also as DA and NA, added to *E. coli* K-12 strain MC 4100 at micromolar concentrations, stimulated cell proliferation and biomass accumulation [21]. 5-HT (1–10 μM) stimulated cell aggregation and microspore formation in the myxobacterium *Polyangium* sp. The opposite effect was induced by 5-HT at concentrations greater than 20 μM, which inhibited cell aggregation and growth of the germinal cell culture [23]. The specific blocker of 5-HT transporter SERT fluoxetine showed a pronounced antimicrobial effect when used at micromolar concentrations in *Pseudomonas aeruginosa*, *Staphylococcus aureus*, and *E. coli* [24].

The new model for the study of transmitter effects was also further elaborated using bacterial bioluminescence. It was found that 5 HT, histamine and DA inhibit bioluminescence at concentrations greater than 80 μg/mL, 100 μg/mL, and 1 mg/mL, respectively. The lower concentrations of these transmitters elicited stimulatory effects. In contrast, NA inhibited luminescence at all concentrations [25]. Putrescine, ACh, taurine and indole (0.1–10 μM) stimulated luminescence of strain *E. coli* K12 TGI containing the lux operon of *Photorhabdus luminescens* ZMI. These compounds moderately attenuated the luminescence of *E. coli* K12 TGI at some higher concentrations [26].

Conversely, the microbiota plays a critical role in regulating host 5-HT. Native spore-forming bacteria *Bacteroides* from the mouse and human microbiota promote 5-HT biosynthesis from enterochromaffin cells (ECs) of the colon, which supply 5-HT to the mucosa, lumen, and circulating platelets. Importantly, microbiota-dependent effects on the transmitters’ levels, i.e., gut 5-HT, significantly influence host physiology and modulate gastrointestinal motility and platelet function [27,28].

Previous studies have demonstrated the existence of several receptors for mammalian hormones such as insulin and neurotensin in bacteria [22,29,30]. In contrast, small molecule transmitters in bacteria act through a specific receptor-signaling system that differs drastically from analogues in higher organisms.

They are two-component signal transduction systems (TCSs) that represent the specific bacterial mechanism of response to external stimuli and control of various processes such as cell division, expression of virulence, quorum sensing (QS), etc. QS is a type of bacterial population response to suprathreshold exposure to external stimuli in the form of release of small diffusible quorum sensing molecules (QSMs) [31].

TCSs consist of a sensory histidine kinase (HK) and a corresponding response regulator (RR). Upon sensing the external signals, HK either activates itself (kinase state) or deactivates itself (phosphatase state), transfer signal to the intracellular domains and finally to the catalytic core that transfer phosphate moiety (P) to the receptor domain of RR. The latter functions as a transcriptional regulator of genes required for the stress response [32,33,34]. For example, in *Citrobacter rodentium*, a murine model for enterohemorrhagic *E. coli* (EHEC), in vivo expressed two-component systems have been identified [33].

Specifically, the QS sensor HK kinase was shown to be a bacterial receptor for the host Adr/NA in EHEC that activates transcription of virulence genes. α-adrenergic antagonists can specifically block the QS response to these signals ([31,35,36]. Two of these bacterial adrenergic receptors have been identified in EHEC as QSMs—QseC and QseE. Through QseC, EHEC activates the expression of metabolic, virulence and stress response genes. Coordination of these responses is achieved by QseC phosphorylating three RRs. The adrenergic sensor QseE phosphorylates the RR QseF, which coordinates the expression of virulence genes involved in the formation of lesions in intestinal epithelia by EHEC, as well as the bacterial stress response SOS [37,38]. The downstream connections of bacterial adrenergic receptors were described in detail in the review by Kendall and Sperandio [39].

In contrast to bacterial adrenergic signaling, 5-HT exerts its physiological effects in EHEC and *C. rodentium* via another membrane-bound histidine sensor kinase, CpxA, which is a bacterial 5-HT receptor. The 5-HT induces dephosphorylation of CpxA, inactivating the transcription factor CpxR, which controls virulence gene expression. Expression of the locus of enterocyte effacement (LEE) is decreased in the presence of physiological concentrations of 5-HT (10 μM and 100 nM), as assessed by quantitative real-time PCR, Western and Northern blots. Increasing intestinal 5-HT by genetic or pharmacological inhibition of SERT decreases pathogenesis, whereas inhibition of 5-HT synthesis increases pathogenesis and decreases host survival [40]. CpxA also recognizes indole [41], a tryptophan-derived bacterial signal [42] that is structurally similar to 5-HT, and has also been shown to decrease LEE gene expression in EHEC [41].

TCS is one of the most important mechanisms of signal transduction in prokaryotes and is present in most bacteria and many archaea [43,44,45]. There are also some TCSs in eukaryotes, such as fungi and plants, albeit in smaller numbers, but none in mammals, where phosphorylation signals are dominated by serine, threonine, and tyrosine kinases [43,46,47,48,49,50]. Conversely, serine/threonine protein kinase domains were recently found in bacterial membrane receptor proteins fused to N-terminal periplasmic sensory domains [51].

In addition to sensory histidine kinases and methyl-accepting chemotaxis proteins, other receptor proteins in bacteria have a similar overall organization, namely with an N-terminal periplasmic or extracytoplasmic sensory domain followed by one or more transmembrane segments and a cytoplasmically located signal transduction domain. This type of organization has been described for membrane-anchored adenylate cyclases, putative diguanylate cyclases and phosphodiesterases, serine/threonine protein kinases, and phosphatases, revealing a much more complex signaling network than was generally assumed prior to genomics [52,53]. Intracellular interaction of transmitters and elements of the cytoskeleton has also been proposed in this or that way [19,26].

Thus, the last decades brought a large body of principally new and important data on the transmitter system in bacteria, of which there was little evidence in the XX century. Such an increase in knowledge about the abundance of transmitter mechanisms in living organisms is changing ideas about the evolution of signaling systems. In particular, it addresses the possibility that the effects of the same transmitter on bacterial cells operate through distinctly different mechanisms—receptor HKs and RRs—than in higher animals.

## 3. Transmitter Systems in Unicellular Organisms

The first data on the presence of transmitters in Protozoa were obtained at virtually the same time as in invertebrate embryos (see below) [54,55,56], although such an idea did not seem very trivial for the time. In particular, 5-HT was found in the flagellated protozoan *Crithidia fasciculata* and the ciliated protozoan *Tetrahymena pyriformis* [55]. Currently, the synthesis of classical transmitters such as 5-HT, ACh, DA, histamine, γ-aminobutyric acid (GABA), etc., has been discovered in various unicellular organisms [57,58,59,60].

Then, the synthesis and release of GABA into the environment by *Paramecium* was detected by HPLC [61]. Catecholamines are naturally synthesized and released by the cells of *T. pyriformis* in the culture medium and progressively accumulate over time, with NA showing the highest level of accumulation [62].

Histamine, 5-HT, Adr, and melatonin stimulated phagocytosis in protozoa [63], while peptide hormones such as adrenocorticotropic hormone (ACTH), insulin, opioids, arginine-vasopressin and atrial natriuretic peptide attenuated it regardless of their chemical structure and functions in multicellular organisms [64]. Elements of the cholinergic system have also been found in protozoa [65].

Regeneration of the ciliary apparatus of *T. pyriformis* is regulated by 5-HT, cAMP, and calcium ions [66]. Binding of the fluorescent D_1_-receptor agonist SKF-38393 by the ciliary protozoan *Tetrahymena thermophile* was specifically inhibited by the addition of an equimolar concentration of a D_1_-antagonist that supports the presence of the dopamine D_1_-receptor here [67]. Catecholamines elicited moderate toxicity in *T. pyriformis* cells, with the most pronounced toxic effect caused by DA and L- DOPA. DA had the greatest effect on inhibiting NA synthesis. Treatment with a higher concentration of DA resulted in a strong excitation of the cells [62].

The swimming behavior of *Paramecium primaurelia* was influenced by the GABA_A_-selective agonist muscimol and was characterized by alternating periods of whirling and forward swimming. This effect was inhibited by the GABA_A_-selective antagonists bicuculline and picrotoxin in a dose-dependent manner. The response to muscimol was also suppressed by nimodipine, a selective antagonist of dihydropyridine-sensitive calcium channels [68].

When the specific GABA_B_ receptor agonist baclofen was used, a dose-dependent inhibition of the duration of ciliary beat reversal induced by membrane depolarization (CR) was observed. This inhibition was antagonized by phaclofen and disappeared after treatment with nifedipine and verapamil. Therefore, these experiments suggest that the GABA_B_ receptor agonist baclofen CR modulates dihydropyridine-sensitive calcium channels through G-protein (G_0_ or G_i_) mediated inhibition [61].

The presence of β-adrenergic [63], and cholinergic [65] sensitivities that control phagocytosis, swimming behavior and cell recognition has also been demonstrated in ciliate protozoa.

The resolution of questions concerning the nature of transmitter receptors in protozoa and prenervous invertebrate embryos has been considerably hampered by the limited extent of molecular biological data to date. Nevertheless, challenging hypotheses have been proposed, and experimental and bioinformatics research is being conducted in this area. For example, a search for an amino acid sequence homologous to the human ACh M_1_- receptor in *Acanthamoeba* failed to find a corresponding ligand-binding site. However, structural bioinformatics revealed the presence of the hypothetical protein L8HIA6, which could be the homolog of human mAChR_1_. This suggestion was supported by positive immunostaining with antibodies against mAChR_1_ [60,69]. 

Fluorescently labelled sites reactive with anti-GABA antibodies were found in the cytoplasm of both immature and non-mating cells of *Paramecium primaurelia*. Colocalization of the anti-GABA antibody reactive sites and of choline acetyl transferase (ChAT, i.e., probably of a presence of ACh) was also detected in this species [57]. The presence and distribution of GABA_A_ receptors in *Paramecium* were detected with monoclonal and polyclonal antibodies. Confocal laser microscopy revealed the expression of α1-, α2-, α3-, α6-, β2/3-, γ2-, ε-, λ-, and θ-subunits of this receptor in *P. primaurelia*. Immunofluorescence is localized in patches distributed on the plasma membrane and mainly in the cytoplasm [57,70]. GABA_A_ α-subunits are retained in an intracellular compartment, presumably the endoplasmic reticulum (ER), by an ER retention signal and wait to be associated with β-subunits [71]. Interaction of α- with β-subunits is thought to result in a conformational change that presumably masks the retention signal and induces transport to the cell surface [70]. The presence of GABA_B_ immune analogs in *Paramecium* has been detected by SDS-PAGE, Western blotting and confocal laser scanning microscopy [61]. Therefore, it was hypothesized that there are transmitter-binding receptors in protozoa [59] from which the signal is transmitted into the cell, deciphered, and executed similarly to cells of multicellular organisms.

Interestingly, the protein composition of neurosecretory vesicles in unicellular organisms is similar to that in higher animals, suggesting a common evolutionary origin. By comparing 28 proteins of the core proteome of neurosecretory vesicles in 13 different species, it was shown that most of the proteins are present in unicellular organisms. For example, synaptobrevin is localized to the vesicle-rich apical and basal pole in the choanoflagellates *Salpingoeca rosetta* and *Monosiga brevicollis* [72]. Thus, this part of the transmitter process could also be similar to that of higher animals.

Protozoa thus exhibit a wide range of transmitter mechanisms comparable to those of higher animals, and surpassing those of fungi, placozoans, and ctenophores in diversity and complexity.

Data on the transmitter systems in fungi are scarce and mainly concern yeast cells. Using highly efficient liquid chromatography, 5-HT, DA and NA were found to accumulate to submicromolar concentrations in *Saccharomyces cerevisiae* cells [73]. Much of the data on this topic comes from applied research on alcoholic beverages and related technologies. 5-HT has been detected in wine and beer [74,75,76]. Intracellular 5-HT was found on days 1–2 but decreased later until the end of alcoholic fermentation (<0.01 ng/mL). Intracellular 5-HT occurrence was previously reported in several *S. cerevisiae* strains (QA23, P24), and the concentration decreased during fermentation [77]. These data suggest that *S. cerevisiae* is capable of synthesizing 5 HT and some of its chemical analogs such as 5-hydroxytryptophan, N-acetylserotonin, 3-indoleacetic acid, and L-tryptophan ethyl ester during alcoholic fermentation of wort [76,78].

*S. cerevisiae* EPF proliferation was stimulated by exogenous transmitters. DA (1 µM) was the most efficient among them, causing an approximately 8-fold growth stimulation. This effect was partially mimicked by apomorphine, a DA receptor agonist. 5 HT and histamine had less significant effects (1.5–2 fold). Similar results were obtained with *Candida guillermondii*, which were also stimulated by 5HT [79]. In contrast, NA had virtually no stimulatory effect on yeast culture growth. These data indicate a specific, apparently receptor-dependent mode of action of transmitters in *S. cerevisiae* cells. Transmitters were accumulated in yeast cells up to (sub)micromolar concentrations without being released into the culture fluid [73]. Apparently, based on general assumptions about the function of transmitters as obligate intercellular messenger systems, the authors concluded that the tested transmitters do not serve as autoregulators in the yeast population [73]. They suggested that transmitters might be involved in the regulation of yeast population development by other ecosystem components, ignoring the fact that these substances might exert “housekeeping” intracellular functions (see “Invertebrate embryos” below).

To date, there are no data on transmitter receptors in fungi, so the mechanisms of transmitter action there remain obscure and much is left to the imagination of researchers.

## 4. Transmitter Systems in Non-Neural Multicellular Invertebrates

Placozoa is the taxon of nerveless metazoans, and data on the transmitters there are exceedingly scanty. Neither 5-HT nor DA has been detected in Placozoa [80]. Moreover, the intracellular glycine (Gly) content in *Trichoplax* has been measured as high as 3 mM, which is significantly higher than other native amino acid transmitter candidates such as L-glutamate (Glu), L-aspartate or GABA. 

The transmitter functions of Glu are not known in placozoans, but Gly at millimolar concentrations (similar to endogenous levels) controls locomotion and contractility and may be a chemoattractant. After 24 h of incubation, 10 mM Gly can induce cytotoxicity and cell dissociation. In contrast, micromolar concentrations (10–100 μM) increased ciliary locomotion in *Trichoplax*, suggesting that Gly may act as an endogenous signaling molecule [80].

The genome of the placozoan *T. adhaerens* [81,82] contains a number of genes encoding proteins for vesicle exocytosis, synapse formation and signal transduction. Accordingly, placozoans possess the most diverse systems of voltage-gated ion channels [83,84,85] and receptors [86]. Notably, at least 13 ionotropic glutamate receptors (iGluRs) have been identified in *Trichoplax* sp. [86]. Both Glu and Gly can be ligands for the different classes of placozoan iGluRs [80], see in detail in [86].

Paradoxically, the transmitter system of placozoans looks much more primitive compared to that of protozoans. The absence of transmitters derived from aromatic amino acids may argue for an independent, possibly degenerative branch of the evolutionary origin of placozoans from other multicellular eukaryotes.

Sponges, as placozoans, are the nerveless group of animals with the primary absence of neural and muscular systems. Recently, however, whole-body RNA sequencing in a sponge (*Spongilla lacustris*) has identified several cell types comprising four major families. These include the family of amoeboid-neuroid cells that express “presynaptic” genes, and another family of cells that express “postsynaptic” genes in the digestive choanocytes, suggesting asymmetric and targeted communication [87]. This suggests that sponge responses are coordinated by alternative integrative systems, including Glu signaling [11,87].

Some sponge classes, such as calcareous sponges, possess multiple ionotropic glutamate receptors (iGluRs) (e.g., *Sycon* and *Leucosolenia*, 5 and 3 iGluRs, respectively) and Homoscleromorpha (18 iGluRs in *Oscarella*). It is suggested that ponge iGluRs, as in mammals, may be controlled by Gly in addition to Glu [86].

In a higher animals, Glu and other anionic transmitters are taken up by SLC17 transporters, monoamines and ACh are taken up by SLC18 transporters, whereas GABA and Gly are accumulated by an SLC32 transporter. An SLC17 transporter with unknown properties is present in choanoflagellates, and some SLC17 transporters are expressed in early branching animals; the above-mentioned SLC18 and SLC32 transporters are also present in the calcareous sponge *Sycon ciliatum* [82]. This suggests that different types of chemicals were available as transmitters very early in the evolution of animals, including those not yet discovered in this or that taxon of primitive animals.

In particular, the presence of 5-HT in sponges was first demonstrated in myocyte-like cells of *S. ciliatum* (Sycettidae, Calcarea) [88]. Using a whole-mount fluorescence technique, a 5-HT-like immunoreactivity was found in the cells of both larval and juvenile *Tedania ignis* [89]. In contrast to GluRs, bioinformatic genomic analysis of four sponge species did not find monoamine receptors. However, electron microscopic studies suggest that monoamine-positive structures are concentrated in the Golgi apparatus of sponge cells [90]. 

Previously, it was shown that sponge contractions can be triggered by the administration of Gly, cAMP, and transmitters Adr and 5-HT, while nitric oxide appears to modulate contraction intensity [91]. The genes for all enzymes of 5-HT and DA synthesis, as well as for transglutaminase, which is required for protein serotonilation, were found in two sponge species. Morphological and biochemical data show the non-canonical intracellular pathway of monoamine action in the functional activity of flagellate sponge cells [90,91].

Since the genome of *Amphimedon queenslandica* contains multiple GABA receptors and at least seven metabotropic Glu receptors, also as of 5-HT and DA [92], it is expected that demosponges are quite capable of signaling and responding. However, this contradicts a later report on bioinformatic genome analysis of four sponge species, which demonstrates the absence of monoamine receptors [90]. At the same time, the genes for all enzymes of 5 HT and DA synthesis and transglutaminase, which are necessary for serotonylation of proteins, were found in two sponge species. These data indicate that sponge cells can realize a non-canonical pathway of monoamine action via post-translational protein modification. Electron microscopic studies suggest that monoamine-positive structures are concentrated in the Golgi apparatus. Morphological and biochemical data reveal the non-canonical intracellular pathway of monoamine action in the functional activity of flagellate sponge cells [90].

The data on the absence of monoaminergic receptors in Porifera, as well as in Placozoa, somewhat contradict the above-mentioned data on transmitter receptors in protozoa and bacteria and obviously require further exhaustive studies. An alternative for receptor regulation in such animals could be the intracellular interaction of transmitters and elements of the cytoskeleton [93,94], for example via protein serotonilation [95].

## 5. Transmitter Systems in Invertebrate Prenervous Embryogenesis

Similar to bacteria and protozoa, transmitter substances such as 5-HT, catecholamines, ACh, and GABA in protozoans only have been detected in the cells of early embryos of all species studied (see [6,96,97,98,99,100,101,102,103,104]).

The transmitters are present in embryos of all taxons since fertilization and the mention of several transmitters for each one in Figure 1 is not the summary of the data from different species, but the reflection of the peculiarities of embryonic cells—the simultaneous presence of such substances in the same cell that radically distinguish them from the majority of cells of adult invertebrate organisms (see [98]).

Eric R. Kandel’s well-known question is: why do neurons have different transmitters, when a single transmitter could mediate all the necessary electrical signals? [105]. Obviously, the point is that the transmission of neural information itself could actually be organized on the basis of a single neurotransmitter, if this were created just at the moment of the formation of the nervous system, especially for the transmission of nervous impulses from one cell to another. However, it is clear from the above that this is not the case.

The likely answer to Kandel’s question is that the diversity of transmitters is due to the fact that their primary function is far different from intercellular signal transmission. The explanation for such a multitransmitter property, based on the ideas of Khachatur Koshtoyantz and further developed by his followers (see [8]), is that neurotransmitter mechanisms are the result of onto- and phylogenetic evolution and transformation of systemcontrolling intracellular synthetic processes. It has been suggested that the primary transmitter function was the regulation of protein synthesis. The diversity of essential amino acids, which are not synthesized in animal cells and are the limiting link in protein synthesis, is the prerequisite for the diversity of signaling substances, which are derivatives of these amino acids. The enzymatic conversion of a small amount of such amino acids into the signal form of the transmitter allows the control of their level in the cells. Consequently, 5-HT is the probe for tryptophan and catecholamines for phenylalanine.

The idea of the conversion of metabolic components into signaling molecules may also be applicable to the formation of acetylcholine, since it is a derivative of the important cell membrane component, phosphatidylcholine, which is derived at least in part from the outer medium choline and consequently requires control of its intracellular levels. The presence of ACh has not yet been directly demonstrated, although several authors have suggested its role in vertebrate oocyte activation [106,107,108]. This could be due to the fact that it is secreted in small amounts and immediately hydrolyzed by the activity of acetylcholine esterase (AChE), which is present in invertebrate eggs [109] and also in spermatozoa [110]. This explains why data on the presence of ACh are mainly limited to those obtained from determinations by the physiological method and experiments on the action of transmitter–receptor ligands. In addition, there is immunofluorescent evidence for the expression of the ACh synthesis enzyme in the sea urchin *Paracentrotus lividus*. Positive immunoreactivity was found in the cytoplasm and on the surface of eggs as well as on the surface of zygotes, suggesting the ability of eggs to synthesize ACh autonomously. Thus, the probability of the presence of ACh in sea urchin embryos was practically proved [111,112].

The specific transmitter of adult mollusks and insects—octopamine—replaces catecholamines in adults of these taxa. Paradoxically, however, no data on the presence of octopamine in insect embryos were found in the literature, although the corresponding receptors and synthetic enzymes are expressed there [113,114,115,116,117,118,119,120,121,122].

Among the signaling substances present in the cells of early embryos, the conjugates of classical transmitters and fatty acids were discovered, which probably also play the role of endogenous signals [123].

As for the formation of transmitter function from such native aliphatic amino acids as Glu, aspartate, and Gly, it has been suggested that such low molecular weight substances were the first transmitters or co-transmitters in evolution [11] and were formed in the cells of primitive organisms by the external flow of such simple molecules. The present mechanism of their transmitter action is probably linked to the electrogenic transport of such amino acids into cells [124].

Prenervous transmitters are usually diffusely distributed throughout the intracellular space, as shown using labelled 5-HT in cells of sea urchin embryos [9]. They are often associated with various vesicular structures, such as 5-HT in almost all blastomeres of the coelenterate *Aurelia aurita* from cleavage until gastrulation [104]. However, 5-HT is asymmetrically localized at the animal pole of the embryo of the mollusk *Tritonia diomedea* [125], while it is concentrated in the prospective cleavage furrow in embryos of the polychaeta *Ophryotrocha labronica* [126].

5-HT was the first transmitter studied in early embryos of sea urchins. Initially, 5-HT levels in *Strongylocentrotus dröbachiensis* embryos were determined by the biological method based on the ability of 5-HT to specifically stimulate the embryonic motility of various nudibranch larvae [6]. The concentrations of 5-HT increase shortly after fertilization, clearly indicating its synthesis in embryonic cells, and then its concentration oscillates, coinciding with cleavage divisions, including the fourth division. Similar results were previously obtained by the same method in early embryos of prostomata—*Lineus desori* (Nemertini) and *Anaitides (Phyllodoce) maculata* (Polychaeta) [127].

Later data obtained by HPLC showed the presence of 5-HT and its methoxy- derivative in both fertilized and unfertilized eggs of *Paracentrotus lividus*. 5-HT appeared to be the predominant form in the unfertilized eggs [128], although there was some doubt about the actual structure of the 5-HT-like substance in the sea urchin egg. Any tryptamine derivative may be present too [129]. Recent experiments also show the possibility of transport and accumulation of exogenous 5-HT in the cells of the early sea urchin embryo *P.lividus*. Administration of the 5-HT precursor 5-hydroxytryptophan in the medium resulted in an increase in intracellular concentrations of 5-HT, which is certainly due to the activity of an intracellular enzyme. Immunohistochemical staining resulted in an increase in 5-HT concentrations in the cytoplasm of embryos in the latter case to 297% compared to the control at the cleavage stage [130]. Thus, the question about the presence of 5-HT in the cells of early embryos of sea urchin should be answered positively.

At the same time, two of the major peaks in the HPLC of *P. lividus* eggs had the same elution time as those of standard A and DA [128] and these catecholamines must be recognized as major transmitter substances of the cleavage divisions of sea urchins.

It is obvious that at least in the gametes of the sea urchin ACh is also present, although this can be judged rather by indirect signs, such as the presence of enzymes for the synthesis of this transmitter [109,110,112].

### 5.1. Molecular Biology of Embryonic Transmitter Signalization

Marine invertebrate embryos are excellent for pharmacological and some biochemical experiments, but offer difficulties for molecular biology approaches. The amount of genomic and transcriptomic data from sea urchins is still very limited. As a result, these marine embryos are far less studied compared to amphibians and mammals.

Nevertheless, the expansion and systematization of knowledge about transmitter mechanisms in adult vertebrates in recent decades became a prerequisite for analogous research in primitive organisms and early embryos. It can be said that researchers of noncanonical transmitter functions have followed in the footsteps of colleagues working with definitive multicellular organisms. They have moved from a purely pharmacological analysis of receptor ligand activity to a direct study of the expression of components of embryonic transmitter mechanisms. This has led to the need to revise established ideas about embryonic transmitter mechanisms (see, e.g., [131]). A more detailed understanding of the processes in early embryos, previously described as serotonergic, requires further investigation of the receptor specificity.

New data provide an opportunity to revise the old main question that was raised at the beginning of research in this field and was even on the title of a review paper—do neurotransmitters function in the same way throughout development? [132]—and received different answers at different times.

In contrast to bacteria and protozoa, much data on the components of transmitter mechanisms in the early embryos of various invertebrates have already been collected using molecular biology approaches. In a number of species, it has been shown that the mRNAs of the components of transmitter mechanisms, including receptors, are the same as of adult organisms. The first publication of this kind was the study of 5-HT_2C_ receptors in the embryonic stages of the nematode *Caenorhabditis elegans* [133]. Later, the amount of such data grew immensely, although even now it is incomplete and fragmentary. 

Of course, mammalian and amphibian embryos have been most thoroughly studied in this context, but there are now numerous papers on echinoderms as well. We must never forget that data on the expression of mRNAs of the components of the transmitter mechanism are indirect evidence for the possible expression of the corresponding protein in early development, but are the prerequisite for it.

The studies on the expression of transmitter mechanisms in echinoderm embryogenesis, where the first and a large number of further experiments in this field were performed, are mainly based on the reading genome of the sea urchin *Strongylocentrotus purpuratus*.

Our molecular biology studies of the expression of components of the sea urchin embryo transmitter system began with a misunderstanding. We made an attempt to detect the expression of 5-HT receptors in *P. lividus* embryogenesis. Four of twenty-six sea urchin genes from *S. purpuratus* determined to be 5-HT receptor homologs in the GNOMON project had homologous sequences in the *P. lividus* nucleotide sequence database—HTR1, HTR2B, HTR4 and HTR of undetermined type. RT-PCR analysis has shown that all 4 genes are expressed in the early developmental stages [134]. However, a more detailed analysis of the amino acid sequences yielded the unexpected result that none of these molecules contains a conservative aspartate residue of the metabotropic 5 HT receptor in the third transmembrane domain [135]. Therefore, the question of 5-HT receptor expression in the early sea urchin embryo remained open for a time [130].

It should be noted that the sequence of the EST clone AM600436, which was obtained from early embryos and shows homology with the hypothetical 5-HT receptor protein LOC589531 of *S. purpuratus*, is in the database https://www.ncbi.nlm.nih.gov/gene (accessed on 13 January 2022) and remains designated as 5-hydroxytryptamine receptor 4 [*S. purpuratus* (purple sea urchin)] genes ID: 589531, updated 13 December 2020, although lacking the conservative aspartate in the 119th position. 

At the same time, the 5-HT_2_-receptor sequence GenBank: CX685095.1, expressed since at least the blastula stage, was present in the ncbi.nlm.nih.gov database [136]. 5-HTR mRNA has been detected in the prism stage of the sea urchin *H. pulcherrimus*, as has the corresponding functional protein [137]. Our preliminary results from the transcriptomes of the sea urchins *S. purpuratus*, *P. lividus* and *Mesocentrotus franciscanus* reveal the presence of mRNA fragments homologous to the genomic sequence of the 5HT_1D_-like receptor, also as D_1_- and D_2_- receptors. This supports our previous data obtained by RT-PCR on the expression of D_2_ receptor mRNAs as well as the specific membrane transporter SERT and the NA transporter NET during cleavage divisions of *P. lividus* cleavage divisions [130].

Transcription of dopamine Hp-DRD_1_ and GABA_A_ receptor (Hp-gabrA) mRNAs has also been detected in developing sea urchin *H. pulcherrimus* since unfertilized eggs and throughout larval development. Expression of the Hp-DRD1 receptor protein itself was detected immunochemically from the rotational blastula stage [138], while the GABA_A_ receptor protein was detected by immunoblotting from the pluteus stage [139].

A large amount of data on embryonic acetylcholine reception has been obtained mainly by Italian scientists. In sea urchin eggs and at subsequent stages of early embryonic development, nicotinic receptors [140] co-localized with AChE activity [109] have been detected. nAChR and mAChR mRNAs and proteins have been detected in spermatozoa and oocytes since fertilization [109,134,141,142]. Muscarinic ACh receptors have been found and localized mainly in the sperm acrosome. This localization might be related to a function in sperm–ovum interaction, for example in regulating the blockade of polyspermy. Nicotinic receptors were found both in the acrosome and in the flagellar membrane, confirming their function in the regulation of sperm propulsion [141]. Later, the presence and localization of nicotinic receptor-like molecules in early embryos of developing *P. lividus* was detected using the specific blocker α-bungarotoxin and the immunoreactivity of the α-7 subunit of the ACh receptor [142,143]. Both methods identify and localize nicotinic receptor-like molecules to sites of active changes in intracellular ion concentration. These are known to lead to either fertilization, sperm propulsion, or coordinated cilia movement [143].

The concept of lack of vesicular transmitter transport in early development and instead passive leakage of transmitter, which arose at the beginning of research on transmitter mechanisms [96], then became a prejudice.

SNARE and members of the Rab protein family are phylogenetically conserved mechanisms involved in the secretory pathway. In particular, this mechanism has been shown to be involved in the cortical reaction in sea urchin egg cells [144,145]. In addition, syntaxin, synaptobrevin (vesicle-associated membrane protein, VAMP), and Rab3 mRNAs and proteins have been detected in early embryos of *P. lividus* and are enriched in regions of the embryo with active secretory functions [146,147,148]. Notably, these proteins are localized in the surface membrane of the interblastomere compartment, suggesting that they may be involved in the processes of blastomere interactions by embryonic transmitters (reviewed in [10]). It has been emphasized that the repertoire of proteins involved in intracellular transport in sea urchin blastomeres is extremely limited; at least syntaxin is present in only one type of protein [148].

Importantly, syntaxin 1A has been shown to interact with several plasma membrane neurotransmitter transporters, including SERT and regulates the transport stoichiometry of the latter. Inhibitors of calcium/calmodulin-dependent kinase II modulate the stoichiometry of 5HT flux and this effect requires syntaxin 1A. The modulation correlates with a change in the affinity of SERT to bind to syntaxin 1A. These data suggest that calcium-mediated signals may serve as triggers for the regulation of syntaxin control of SERT conducting states [149].

The SM family (sec1/muc18) or syntaxin-binding protein is also part of the SNARE complex. It has been identified in sea urchins in association with exocytosis of cortical granules during fertilization, in cleavage furrow membranes, and in secretory cells later in development. In oocytes and eggs of two sea urchin species, *Lytechinus variegatus* and *S. purpuratus*, sec1/munc18 is localized in the plasma membrane and cortical region of the egg. The protein is also expressed and enriched in the membranes of the cleavage furrow in the early embryo. It colocalizes with its cognate binding partners syntaxin and Rab3 in high molecular weight complexes, suggesting that the exocytotic machinery functions as a multiprotein subunit to mediate regulated secretion in sea urchins [150].

### 5.2. Physiological Effects of the Transmitters

The extensive arsenal of transmitter components described above is the basis for a variety of their regulatory influences on embryonic development.

The first event in sea urchin embryonic development involving transmitters is fertilization, in which ACh plays a pivotal role. Exposure of egg cells to 1 mM ACh + 1 μM eserine prior to fertilization resulted in incomplete membrane depolarization and consequently enhanced polyspermy, whereas lower concentrations of ACh caused further developmental abnormalities, as well as 0.045 AChE Units/mL [112]. In sea urchin fertilization the acrosome reaction may be blocked by curare and α-bungarotoxin [151].

Exposure of sea urchin eggs to nicotine leads to polyspermy at fertilization in a dose-dependent manner. Unexpectedly, this effect of nicotine is mediated by some non-cholinergic signaling pathways. In contrast to ACh and carbachol, nicotine induces a dramatic restructuring of sea urchin cortex microfilaments via a direct acceleration of the polymerization kinetics of G-actin and attenuates the depolymerization of pre-assembled F-actin [152].

After fertilization, ACh and carbachol trigger an intracellular Ca^2+^ increase in two sea urchin species: *P. lividus* and *L. pictus*, which was partially inhibited by atropine. Exposure to ACh receptor agonists after fertilization resulted in transient changes in chromatin structure. It is hypothesized that muscarinic receptors may be involved in the (presumably Ca^2+^-dependent) modulation of nuclear status during the first cell cycle [153].

It was suggested that the cholinergic system may be involved in two distinct developmental processes in which a particular type of ACh receptor is active during a specific time window. The first function, which occurs during fertilization, is the result of autonomously synthesized ACh in spermatozoa, whereas the second function, which occurs after fertilization, is due to maternal ChAT molecules that are assembled on the oolemma along with the egg maturation and fertilization processes [112].

In parallel with cholinergic regulation of fertilization events, sperm–ovum fusion triggers a signaling cascade that releases intracellular calcium (Ca^2+^) from the endoplasmic reticulum (ER). In sea urchins, the large Ca^2+^ transient is controlled by two distinct pathways—the production of inositol-1,4,5-triphosphate, which triggers the initial phase of Ca^2+^ release, and the production of nitric oxide (NO), which maintains the duration of the Ca^2+^ wave. The sea urchin homologue of the G protein-coupled receptor for histamine (suH1R), which activates the production of NO, was found on the egg surface. Histamine treatment causes fluctuations in the resting levels of NO in the egg, while H_1_R-specific antagonists or antibodies inhibit the increase in NO normally observed at fertilization [154].

Most of the embryophysiological and pharmacological data on transmitter functions in further prenervous embryogenesis was collected in early sea urchin embryos and are described in the monographs of Prof. Gennady A. Buznikov [96,98] and some reviews [8,10,99,132,142,155]). This object offered a number of advantages at the initial stage of research, which allowed to obtain maximum data on various aspects of the functions of transmitter mechanisms in early embryogenesis. The high speed of development of transparent, i.e., easily visually controllable, numerous and genetically uniform sea urchin embryos allowed the creation of experimental models that offered exclusive experimental opportunities.

Experiments in which sea urchin cleavage divisions were blocked and restored under the action of transmitter antagonists and agonists accordingly proved to be the simplest, most easily quantifiable, and most fruitful in the early stages of development [96,98,130,131,156,157]. In the first phase of these experiments, researchers felt that the indole derivatives had the most pronounced embryostatic effects. This led to the hasty conclusion that they all had serotonergic nature [98].

It should be noted that the phenomenon of blockade of early development under the action of transmitter antagonists may be due to a number of factors affecting the various regulatory cascades and cellular elements. They include, at least, fertilization and cell cycle initiation ([96,142,154,158] actin cytoskeleton control [159], for example, via 5 HT_2A_-receptor and the microtubule-associated protein MAP1A link [160], and regulation of contractile ring assembly [161].

Apparently similar results, namely division arrest, may be caused by exactly opposite actions of the ligands of the 5-HT and catecholamine receptors in the embryonic cytoskeleton. Antagonists of 5-HT have been shown to arrest development by decreasing the stiffness of the cytocortex, whereas catecholamine antagonists significantly increase it and also cause developmental arrest [159].

DA and GTP elicit stimulation of adenylate cyclase activity in *P. lividus* that exceeds the effects of Adr and NA [162]. The agonist of the D_1_-receptor SKF-38393 stimulated adenylate cyclase activity, whereas the two D_1_ dopamine antagonists, SCH -23390 and SKF-83566, suppressed the stimulatory effect of dopamine. In addition, the D_2_ dopamine agonists produced a dose-dependent inhibition of dopamine-stimulated adenylate cyclase activity. Thus, adenylate cyclase is dopaminergically regulated in sea urchin eggs via dopamine receptors that have properties similar to those in the mammalian brain [163]. Moreover, recent experiments in cleaving *P. lividus* embryos have shown that DA-receptor ligands induce specific damage to the tubuline cytoskeleton that is distinct from that induced by 5 HT-antagonists [131]. This confirms the difference in the signaling pathways of these two transmitters in these cells.

### 5.3. Intracellular Transmitter Activity

It was discovered quite a long time ago that that the embryostatic effects of transmitter-receptor antagonists in sea urchin embryos were as pronounced as the degree of their lipophilicity and, accordingly, the ability to penetrate within the cells [164,165,166]. On this basis, the intracellular localization of functionally active embryonic transmitter receptors was proposed, which was ideologically linked to the above-mentioned concept of Koshtoyanz and Buznikov about primarily intracellular transmitter functions. Direct experimental evidence of intracellular localization of transmitter receptors in sea urchin embryos was provided for *H. pulcherrimus* DA receptors by immunohistochemistry [138]. Intracellular localization of transmitter receptors has also been demonstrated in protozoans [67], amphibian embryos [167] and mammalian embryos [168], see for review [155].

It has been suggested that transmitter receptors synthesized at the inner surface of the endoplasmic reticulum in definitive organisms and then transported to the plasma membrane of the cell may also be active at intracellular membranes [10]. The uniformly distributed activity in the cytoplasm of adenylate cyclase localized in the endoplasmic reticulum of fertilized sea urchin eggs supports this possibility [169]. This suggestion is also supported by data on the protective effect of dibutyryl derivatives of cyclic nucleotides against the embryostatic effect of 5-HT antagonists [167]. Later, data were obtained on the activity of adenylate cyclase in association with the sarcoplasmic reticulum, nuclear envelope, and other internal membranes in cardiocytes [170]. The presence of such a functional structure “receptorosome” could explain the relatively high concentrations of receptor ligands that can elicit physiological responses compared to classical membrane receptor responses.

It was originally thought that the intracellular localization of functional transmitter receptors might be an exclusive feature of embryogenesis. However, intracellular transmitter reception (or more cautiously specific binding) has also been discovered in cells of definitive organisms: histamine reception in rat liver cells [171,172], 5-HT_7_-receptors in the developing rat brain [173], mAChR in mouse neuroblastoma cells [174] and mollusk neurons [175,176,177]. Prou et al. [178] noting a predominant intracellular localization of D_2_-receptors in COS-7 and HeLa cell cultures associated with the endoplasmic reticulum, hypothesized that D_2_-receptors might play a role in intracellular compartments. Thus, the interaction of D_2_-receptors with heterotrimeric G proteins in the ER and Golgi could be a regulatory factor for the secretion of cell products [179].

### 5.4. Intercellular Transmitter Signaling

Among intracellular localization the canonical one of embryonic transmitter receptors at the plasma membrane of oocytes [98] and blastomeres [180] of echinoderms has been demonstrated using pharmacological, embryophysiological, electrophysiological, and ligand-binding approaches. 5-HT receptors located on the cell membrane of sea urchin embryos are involved in oocyte maturation [98], control of free intracellular Ca^2+^ levels [161], and direct signal exchange between blastomeres [10,180,181].

In this case, the effects were demonstrated not only for 5-HT antagonists, but also for this transmitter itself and its agonists. The latter mimic the interblastomere signal that determines the prospective fate of blastomeres, while the 5-HT receptor antagonists mimic the elimination of the interblastomere signal [10,181,182,183]. In so-called “micromere model” [184], the ligands, which poorly penetrate the cell membrane, were sufficiently more effective in this regard than their lipophilic analogues, which readily penetrate the cells of sea urchin embryos [180].

Based on these data, the “protosynapse” hypothesis was formulated on the bilateral symmetrical structure, in which both sister blastomeres are the source and target of the signal by the transmitter molecules, which is transported into the intercellular compartment by a highly conservative SNARE transport system. Originally, it might be necessary to avoid highly active signaling molecules from the intracellular space to limit their effect. Using whole-cell patch clump, 5 HT and its agonists were shown to elicit specific inward currents in cleaving *P. lividus* embryos. Shorter latent periods and higher amplitude of responses when the 5-HT agonists are applied into the cleavage furrow compared to the free surface of the blastomeres indicate that the 5-HT receptor is likely located in the contact zone of the blastomeres [183,185].

“Post-division adhesion” of blastomeres [186] limits the leakage of the transmitter from the interblastomere cleft into the outer medium, creating blastomere asymmetry and predetermining its prospective fate [10,180,184].

At later stages of development, such membrane receptors may participate in the formation (or regeneration) of ciliary motility ([137,138,139,187] and in the control of morphogenetic cell movements during gastrulation [188,189], as well as in a variety of processes in the later stages of ontogeny in which neurotransmission occupies an honorable but not unique place (see [155]).

## 6. Conclusions

The most important components of transmitter mechanisms, the transmitter substances themselves, are certainly present in bacteria, fungi, and protists. Moreover, more than one transmitter may be present in all these types of cells. Transmitters and their analogs have a number of physiological effects on the processes of cell division and social behavior in bacteria, fungi, and protozoa. At the same time, the question of the presence of the second most important component of the transmitter mechanism—functional receptors and corresponding signal transduction pathways—has not yet been significantly and unambiguously resolved. These data are still sparse, in low numbers, or, as in the case of fungi, not available at all. Nevertheless, their presence must be considered established, at least in the protozoa, even if it is somewhat contradicted by their absence in the more highly organized Porifera and Placozoa. However, if the existence of specific receptors in fungi and placozoans cannot be further proved, there remains the alternative possibility of direct interaction of the transmitters with the contractile elements of the cytoskeleton and some other proteins.

5 HT, catecholamines, GABA, and ACh are also found in the cells of prenervous embryos of invertebrate animals. As in primitive organisms, more than one transmitter may be present in the same cell at the prenervous stage of embryo development.

Finally, molecular biology studies have led to extensive new insights into the expression of various components of transmitter mechanisms: receptors, transporters, and enzymes for synthesis and degradation. Nevertheless, a limited number of RT-PCR and immunological studies suggest that enzymes of transmitter synthesis are present in sea urchin embryogenesis, as are a number of transmitter receptors, including GABA-, DA-, 5-HT-, and ACh-, which are identical with those of adult organisms.

Of course, the presence of mRNA expression does not automatically imply, but is a prerequisite for, the expression of the corresponding proteins. Depending on the transmitters established in early embryonic cells, several transduction pathways of transmitter signals may be active in the same cell. The transmitter system may be complete, i.e., it may contain the entire set of enzymes and transport components. For example, the expression of mRNAs of D_1_- and D_2_-receptors, n- and m-AChR, and 5-HT_2_ and GABA_A_ receptors was detected in different embryonic stages of *P. lividus*. 

Transmitters have been shown to be involved in the regulation of the cell cycle in its initiation, karyokinesis and cytokinesis, but also in a processes of signal exchange between blastomeres, etc., by activating receptors localized both intracellularly and at the surface membrane also as corresponding signal transduction pathways (Figure 2). 

It should be noted that even a cursory analysis of the publications on the physiological effects of transmitters and other neuropharmaca shows that most of them date from the second half of the twentieth century. This is natural, because at that time pharmacological experiments were the simplest and most accessible method of studying the function of transmitters in prenervous embryogenesis. The new century has produced overwhelming data on the molecular biological features of the process under consideration, leading to a certain disharmony in the development of these studies and urgently calling for a revision and reconsideration of the previous embryophysiological data at a new methodological level.

In particular, a retrospective evaluation of these data [96] must take into account that—firstly—in recent decades the classification of transmitter receptors has been completely changed and—secondly—all neuropharmaca used in these experiments have been characterized in mammalian cells and are therefore less useful for sea urchin embryos, especially with respect to the serotonergic specificity of the ligands. Maximal embryostatic activity in sea urchin embryos was found with 5-HT_1_ receptor antagonists [130], although the significant similarity of sea urchin 5-HT_1D_ and D_1_ receptors requires a cautious approach to the results of previous studies proposing synthetic indole derivatives as likely 5-HT receptor antagonists [96]. Our data on the different effects of DA- and 5-HT-antagonists, especially on the tubulin cytoskeleton [131], on the one hand, and the different protector spectra [130], on the other hand, clearly show the difference in signal transduction pathways of dopaminergic and serotonergic ligands, especially their receptor links. It is obvious that further, especially immunological, studies of the expression of 5-HT- and DA-receptors in early embryogenesis of sea urchins would allow to solve the problem of transmitter specificity of the processes of embryonic regulation and the interaction of different transmitter systems.

The expression of transmitter receptors, SERT, NET and components of the SNARE complex, also as enzymes of transmitter synthesis and degradation, improves our understanding of prenervous transmitter processes. We can demonstrate the presence of a complete set of the major components of transmitter mechanisms identical to those of adult organisms and their functional activity in a various non-nervous organisms and prenervous embryos. Nevertheless, it is worthwhile to search for unknown components of the embryonic transmitter system that are likely to be transiently expressed.

The overall picture that emerges from the above looks illogical and rich in evolutionary leaps in some cases: Bacteria synthesize 5-HT and evolve a specific receptor mechanism, but placozoans have neither. 5-HT has been found in sponges, but the data on the presence of the corresponding receptors are controversial. The main reason for this is probably the lack of data in this area, which is still very sparse compared to analogous data on neuronal or even embryonic processes in invertebrates. Possibly, filling such gaps would save our understanding of transmitter systems from such controversy and missing receptors could finally be found. On the other hand, the evolution of transmitter systems has promised no one to be easy on the minds of researchers.

## Figures and Tables

**Figure 1 biomolecules-12-00271-f001:**
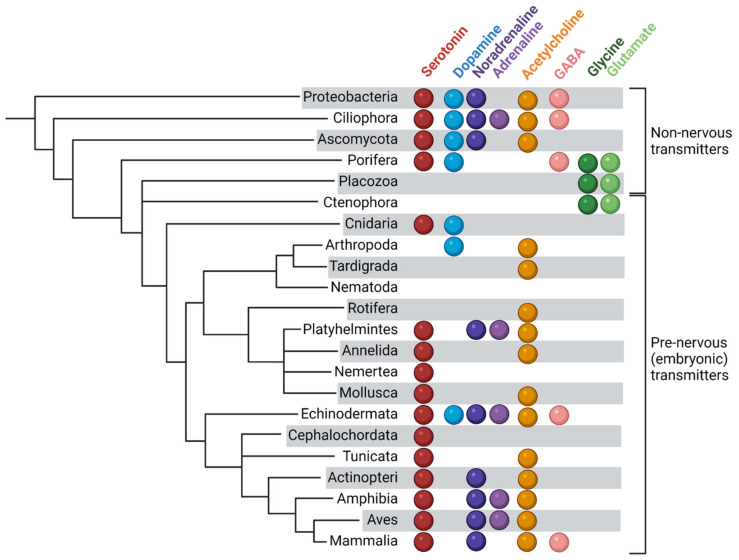
Transmitters in prenervous embryos and in organisms having no nervous system. An almost complete diversity of transmitters has developed already in unicellular and non-nervous multicellular organisms.

**Figure 2 biomolecules-12-00271-f002:**
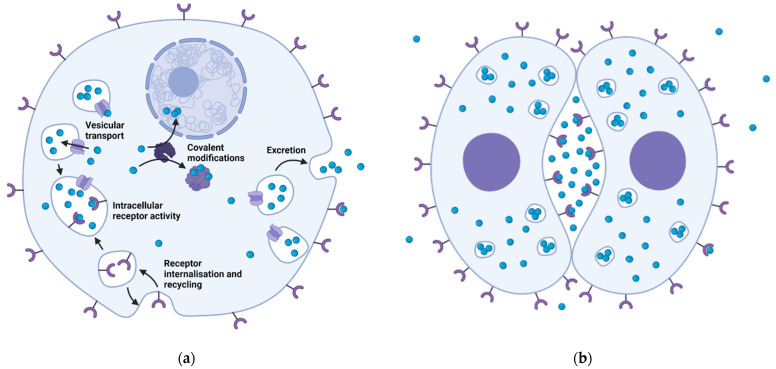
Classical and non-canonical transmitter mechanisms in embryonic cells. (**a**). Possible mechanisms of intracellular activity of transmitters. Receptor internalization and vesicular transport activity may hypothetically provide for the intracellular activity of receptors in the composition of “receptorosomes.” In addition, monoamines are substrates for covalent modification of proteins and influence their activity. (**b**). The concept of protosynapse. The concentration of secreted transmitter in the interblastomeric space provides activation of receptors on interblastomeric surfaces and ensures polarization of the cell with respect to the external/internal environment.

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
