# Peer review of "Non-Neuronal Transmitter Systems in Bacteria, Non-Nervous Eukaryotes, and Invertebrate Embryos"

_biomolecules, 2022, doi:10.3390/biom12020271_

Round 1

Reviewer 1 Report

The review manuscript by Shmukler and Nikishin discusses transmitter systems in non-neuronal cells. The topic is timely and of importance and likely of interest to the readership of Biomolecules. The manuscript is generally well-organized and -written. My comments, given below in the order they appear in the manuscript, are mostly minor and concern presentation and clarification.

- Introduction. A general comment on when discussing antagonists of transmitter receptors. It should be made clear, preferably early on, that an antagonist of a vertebrate transmitter-gated ion channel (e.g., a GABA-A or a nicotinic receptor) should not necessarily be expected to inhibit the actions of a transmitter in a non-neuronal cell. Lack of effect could be due to a bacterial (or sea urchin) analog differing in structure and specificity from the vertebrate/neuronal receptor. Alternatively, the bacterial "receptor" could be of wholly different type and thus not sensitive to an inhibitor of vertebrate/neuronal receptor.

- Introduction, "neurotransmitters have nothing to do within embryos". This should be qualified with "personal recollection" if either author was present, or a verifiable reference. Otherwise, it is hearsay and thus not appropriate in a scientific article.

- 2. Transmitter system.. Change to "systems"?

- Please pay attention to italicize all scientific names of species.

- 2, "suggest that bacterial..". Perhaps replace with "indicated that bacterial..".

- 2. When discussing the physiological effects of transmitters in bacteria, make sure to specify that the transmitters were applied extracellularly to the medium. Also, if the targets of transmitters are located in the cell, then make clear that cellular uptake or diffusion into the cell plays a role in apparent effectiveness.

- 2, last paragraph. Perhaps replace "corpus" with "body".

- 3, "GABA-related molecules". Unclear. GABA and what else?

- 4, "endogenous Gly". Do you mean intracellular Gly?

- 4, "Some sponge classes, such as..". The sentence is unclear (some words may be missing or parentheses placed incorrectly).

- 4. What is "motif organization"? Same sentence: glycine is a regular co-agonist of a mammalian NMDAR, it is then unclear why it needs to be pointed out separately for sponge iGluR.

- 4, "Since the genome of..". Is it multiple GABA-A or GABA-B receptors?

- 5, "Kandel's well-known..". Provide a full name for Dr. Kandel.

- 5, "cell membrane component, choline..". Of course, choline is not a component of membrane, but rather a component of phosphatidylcholine. Re-word.

- 5.1, "mRNAs .. are highly homologous to..". Unclear why would an mRNA of a receptor in an embryo not be identical to the mRNA of the receptor in adult organism? Is the RNA-editing mechanism different? Or did you mean the distribution/presence of various mRNAs?

- 5.1. Unclear what is meant by "choline reception". Clarify.

- 5.2. Unclear why concentrations of ACh are expressed in ACh esterase units. Same paragraph: curare and a-bungarotxin also bind to the muscle-type nicotinic receptor (not just the neuronal a7).

- 5.2, next paragraph. Unexpectedly and probably in the same sentence is awkward. Re-word.

- 5.2. Carbachol is also a nicotinic ACh receptor agonist.

- 5.3. Unclear in what way the finding that adenylate cyclase is uniformly distributed in ER supports the idea that transmitter-gated ion channels are active at intracellular membranes.  Also, "functionally active" seems somewhat redundant (either functional or active should be sufficient).

Author Response

Thanks for careful review. Please, see answers below.

Introduction. A general comment on when discussing antagonists of transmitter receptors. It should be made clear, preferably early on, that an antagonist of a vertebrate transmitter-gated ion channel (e.g., a GABA-A or a nicotinic receptor) should not necessarily be expected to inhibit the actions of a transmitter in a non-neuronal cell. Lack of effect could be due to a bacterial (or sea urchin) analog differing in structure and specificity from the vertebrate/neuronal receptor. Alternatively, the bacterial "receptor" could be of wholly different type and thus not sensitive to an inhibitor of vertebrate/neuronal receptor.

Starting from line 90 of corrected text the insertion made as follows:

Here and below, it should be taken into account that the specificity of the ligands of mammalian transmitter receptors differs significantly from that of non-nervous organisms and prenervous embryos due to very significant differences in the structure of receptors. This is especially true for bacteria, where the nature of transmitter receptors is fundamentally different from that of eukaryotes. In addition, the ligands of transmitter-gated ion channels, in principle, affect the processes of cell division to a much lesser extent than the ligands of metabotropic G-protein coupled receptors.

- Introduction, "neurotransmitters have nothing to do within embryos". This should be qualified with "personal recollection" if either author was present, or a verifiable reference. Otherwise, it is hearsay and thus not appropriate in a scientific article.

The recollection of a speech from a colleague at academic council belongs to Prof. Buznikov former scientific advisor of one of the authors of the review. The text has been removed.

- 2. Transmitter system.. Change to "systems"?

Corrected to "systems".

- Please pay attention to italicize all scientific names of species.

The text is carefully proofread and all species names are given in italics.

- 2, "suggest that bacterial..". Perhaps replace with "indicated that bacterial..".

replaced.

- 2. When discussing the physiological effects of transmitters in bacteria, make sure to specify that the transmitters were applied extracellularly to the medium. Also, if the targets of transmitters are located in the cell, then make clear that cellular uptake or diffusion into the cell plays a role in apparent effectiveness.

– Added by “into the medium” in line 81. Similar correction made in lines 91-92

- 2, last paragraph. Perhaps replace "corpus" with "body".

Replaced.

- 3, "GABA-related molecules". Unclear. GABA and what else?

Corrected: Colocalization of the anti-GABA antibody reactive sites

- 4, "endogenous Gly". Do you mean intracellular Gly?

Replaced by “intracellular”

- 4, "Some sponge classes, such as..". The sentence is unclear (some words may be missing or parentheses placed incorrectly).

Sentence revised: Some sponges possess multiple ionotropic glutamate receptors (iGluRs), e.g. calcareous sponges Sycon and Leucosolenia have 5 and 3 iGluRs, respectively, and Oscarella (Homoscleromorpha) have18 iGluRs.

- 4. What is "motif organization"? Same sentence: glycine is a regular co-agonist of a mammalian NMDAR, it is then unclear why it needs to be pointed out separately for sponge iGluR.

Since the sensitivity of the receptors of lower animals can differ significantly from those of mammals, as is the case, in the embryos of sea urchins, we considered it appropriate to mention the ability of glycine to interact with sponge glutamate receptors too. The text has been revised: It is suggested that sponge iGluRs, as in mammals, may be controlled by Gly too.

- 4, "Since the genome of..". Is it multiple GABA-A or GABA-B receptors?

Authors quoted GABA B receptor, 1 and GABA B receptor, 2

- 5, "Kandel's well-known..". Provide a full name for Dr. Kandel.

Full name of Dr Eric R. Kandel added.

- 5, "cell membrane component, choline..". Of course, choline is not a component of membrane, but rather a component of phosphatidylcholine. Re-word.

Sorry for mistake. The sentences revised as follows: acetylcholine, since it is a derivative of the important cell membrane component, phosphatidylcholine, which is derived at least in part from outer medium choline and consequently requires control of its intracellular levels

- 5.1, "mRNAs .. are highly homologous to..". Unclear why would an mRNA of a receptor in an embryo not be identical to the mRNA of the receptor in adult organism? Is the RNA-editing mechanism different? Or did you mean the distribution/presence of various mRNAs?

This phrase is an echo of a long discussion on the similarity or difference in the receptor mechanisms of prenervous embryos and adult organisms. No wonder in many of the works of Prof. Buznikov and his collaborators made the reservation "embryonic receptors or their functional analogues". In fact, this question was included in the title of the key review by Buznikov et al. (1996) From oocyte to neuron: do neurotransmitters function in the same way throughout development? Cell. Molec. Neurobiol. 1996, 16, 532-559 The issue of the unity of embryonic and definitive transmitter receptors at the mRNA level has been resolved, but not yet completely, quite recently, and we considered it appropriate to emphasize this.

- 5.1. Unclear what is meant by "choline reception". Clarify.

Sure, acetylcholine reception. Corrected

- 5.2. Unclear why concentrations of ACh are expressed in ACh esterase units. Same paragraph: curare and a-bungarotxin also bind to the muscle-type nicotinic receptor (not just the neuronal a7).

ACh concentration is not measured in ACh esterase units. The text quotes that low concentrations of ACh act in the same way as AChE at the mentioned concentration. Further, the mention of the binding of bungarotoxin to the a7 subunit of n-AChR has been eliminated.

- 5.2, next paragraph. Unexpectedly and probably in the same sentence is awkward. Re-word.

“Probably” avoided

- 5.2. Carbachol is also a nicotinic ACh receptor agonist.

Shorted to “After fertilization, ACh and carbachol trigger an intracellular Ca2+ increase”

- 5.3. Unclear in what way the finding that adenylate cyclase is uniformly distributed in ER supports the idea that transmitter-gated ion channels are active at intracellular membranes.  Also, "functionally active" seems somewhat redundant (either functional or active should be sufficient).

In this case, we are not talking about transmitter-gated ion channels, but, on the contrary, about metabotropic G-protein coupled transmitter receptors. The activity of adenylate cyclase associated with ER indirectly testifies in favor of the possible intracellular localization of functional transmitter receptors.

"functionally active" is a tracing-paper from the stable Russian phrase used in this case. Shortened in text to “active”

Reviewer 2 Report

This review summarizes current knowledge about the role of classical neurotransmitters in bacteria, protozoa, non-neural multicellular invertebrates and  prenervous  invertebrate embryos.  It is an interesting and highly informative review which would be of interest for researchers working in the field of invertebrate physiology as well as neurobiology.

Major Comments:

The review is focused on transmitters such as 5-hydroxytryptamine, acetylcholine and catecholamines,  and their receptors, transporters and enzymes for synthesis and degradation. However, information about neurotransmitters such as GABA  and glutamate seems to be insufficient. For example,  GABA-synthesizing enzyme glutamic acid decarboxylase (GAD) is mentioned nowhere in the text.

The text is relatively wordy and transmitter systems are not described in all chapters in a defined order. The  authors should  consider  the possibility to include a table for  various transmitters, or components of transmitter mechanisms, in various organisms, which could contain related  seminal references.

Another problem of this review is missing list of abbreviations. There are too many abbreviations which makes  the reading of the text difficult. Some abbreviation are not explained and/or  seem to be not necessary or superfluous.

Specific Comments:

List of Contents and  List of abbreviation are missing.

Line 96: abbreviation “SERT” is not explained

Line 117: the term „two-component signal transduction systems (TCSs) „ seems to be very universal . How  specific is this system in invertebrate physiology?

Line 122: „...response regulator (RR)“, another universal term.  How important is this abbreviation and what is known about this protein ?

Line 125: abbreviation  „EHEC“ is not explained when used for the first time, it is explained only later (line 137)

Line 125:  „26“ should be removed

Line 132: „ HAMP, PAS or GAF“ abbreviations not explained

Line 133: abbreviation „receptor domain (REC)“ seems  not to be necessary, i tis not used latter

Line 137: „A“ abbreviation not explained

Line 140: „QseC and QseE“ abbreviations not explained

Line 144: „SOS“ abbreviation is used only 1x

Line 298: „iGluRs“ abbreviation not explained

Line 317: „In animals,…“, please, specify the organisms

Line 317: „SCL“ not explained

Lines 357-358: „Similar to bacteria and protozoa, transmitter substances such as 5-HT, catecholamines and GABA have been detected in the cells of early embryos ….“ . GABA is not mentioned  in the chapter 2.

Fig.1: This scheme does not contain GABA and glutamate which are the most important neurotransmitters  in vertebrates and thus are of  interest for readers in the field of neuroscience.

Fig.2: Panels "a" and "b" are not indicated in this figure

Author Response

Thanks for careful review. Please, answers see below.

Major Comments:

The review is focused on transmitters such as 5-hydroxytryptamine, acetylcholine and catecholamines,  and their receptors, transporters and enzymes for synthesis and degradation. However, information about neurotransmitters such as GABA  and glutamate seems to be insufficient. For example,  GABA-synthesizing enzyme glutamic acid decarboxylase (GAD) is mentioned nowhere in the text.

The text is relatively wordy and transmitter systems are not described in all chapters in a defined order. The  authors should  consider  the possibility to include a table for  various transmitters, or components of transmitter mechanisms, in various organisms, which could contain related  seminal references.

Figure 1 exhaustively represents the distribution of the transmitters. Transmitter receptors and other components of this system, where found, are described in the text, but these data are too sketchy to provide a more or less uniform table. In addition, it would be too cumbersome and difficult to understand due to the heterogeneity of the material. In addition, preparing such a table would require quite a long time, despite the fact that 5 days are allotted for a “minor revision”.

Another problem of this review is missing list of abbreviations. There are too many abbreviations which makes  the reading of the text difficult. Some abbreviation are not explained and/or  seem to be not necessary or superfluous.

We didn't find a place for List of Contents in the article submission form. If needed, it can be easily formed from overview section headings as follows:

  1. Introduction
  2. Transmitter systems in bacteria
  3. Transmitter systems in unicellular organisms
  4. Transmitter systems in non-neural multicellular invertebrates
  5. Transmitter systems in invertebrate prenervous embryogenesis.
    • Molecular biology of embryonic transmitter signalization
    • Physiological effects of the transmitters
    • Intracellular transmitter activity
    • Intercellular transmitter signaling
  6. Conclusions

References

Specific Comments:

List of Contents and  List of abbreviation are missing.

We did not find anywhere in the Rules for the preparation of publication a place for a list of abbreviations. However, we are ready to provide it by placing it, if necessary, after the keywords.

List of abbreviations

5-HT - 5-hydroxytryptamine (serotonin)

ACh – acetylcholine

AChE - acetylcholine esterase

Adr – adrenaline

ChAT - choline acetyl transferase

DA – dopamine

DOPA - l-3,4-dihydroxyphenylalanine

DRD – dopamine receptor

EHEC – enterohemorrhagic E. Coli

ER – endoplasmic reticulum

GABA - γ-aminobutyric acid

Glu - L-glutamate

Gly – glycine

HK - histidine kinase

HPLC – high performance liquid chromatography

HТR – 5-HT-receptor

iGluR - ionotropic glutamate receptors

LEE - locus of enterocyte effacement

mAChR – muscarinic ACh-receptor

NA – noradrenaline

nAChR – nicotinic ACh-receptor

NET – noradrenaline transporter

QSM - quorum sensing molecule

RR - bacterial response regulator

RT-PCR - Reverse transcription polymerase chain reaction

SERT – 5-HT-transporter

SLC - Solute Carrier

SNARE – Soluble N-ethylmaleimide-Sensitive Factor Attachment Proteins receptor

suH1R - sea urchin histamine receptor 1

TCS – bacterial two-component signal transduction system

Line 96: abbreviation “SERT” is not explained

SERT – is a well-established abbreviation for serotonin transporter  or SLC6A4

Line 117: the term „two-component signal transduction systems (TCSs) „ seems to be very universal . How  specific is this system in invertebrate physiology?

To our knowledge TCS is the standard term for bacterial sensor system based on histidine-kinases

Line 122: „...response regulator (RR)“, another universal term.  How important is this abbreviation and what is known about this protein ?

RR is standard component of TCS and routine term for it

Line 125: abbreviation  „EHEC“ is not explained when used for the first time, it is explained only later (line 137)

Transfered

Line 125:  „26“ should be removed

 - removed

Line 132: „ HAMP, PAS or GAF“ abbreviations not explained.

Redundant information in this case. Eliminated

Line 133: abbreviation „receptor domain (REC)“ seems  not to be necessary, i tis not used latter

Eliminated

Line 137: „A“ abbreviation not explained

Corrected to “Adr” – adrenaline and added to list of abbreviations

Line 140: „QseC and QseE“ abbreviations not explained

Reference to above mentioned abbreviation QSMs (now line 130) is inserted

Line 144: „SOS“ abbreviation is used only 1x

"SOS" (the international distress signal) is a name for the widely accepted concept of a bacterial global response to DNA damage in which the cell cycle is arrested and DNA repair and mutagenesis are induced

Line 298: „iGluRs“ abbreviation not explained

Decipher of “iGluRs” - ionotropic glutamate receptors – added (now line 309)

Line 317: „In animals,…“, please, specify the organisms

Reworded: In a higher animals

Line 317: „SCL“ not explained

SCL is a common abbreviated name for a transporters

Lines 357-358: „Similar to bacteria and protozoa, transmitter substances such as 5-HT, catecholamines and GABA have been detected in the cells of early embryos ….“ . GABA is not mentioned  in the chapter 2.

Sentence clarified: Similar to bacteria and protozoa, transmitter substances such as 5-HT, catecholamines, ACh, and GABA in protozoans only have been detected in the cells of early embryos of all species studied

Fig.1: This scheme does not contain GABA and glutamate which are the most important neurotransmitters  in vertebrates and thus are of  interest for readers in the field of neuroscience.

Data on GABA are inserted in the Figure. Data concerning glutamate are too limited.

Fig.2: Panels "a" and "b" are not indicated in this figure

Indications of panels a. and b. are added